# Coronary Angiography Upgraded by Imaging Post-Processing: Present and Future Directions

**DOI:** 10.3390/diagnostics13111978

**Published:** 2023-06-05

**Authors:** Benoit Caullery, Laurent Riou, Gilles Barone-Rochette

**Affiliations:** 1Department of Cardiology, University Hospital, 38000 Grenoble, France; bcaullery@chu-grenoble.fr; 2University Grenoble Alpes, INSERM, CHU Grenoble Alpes, LRB, 38000 Grenoble, France; laurent.riou@univ-grenoble-alpes.fr; 3French Clinical Research Infrastructure Network, 75018 Paris, France

**Keywords:** CASS vFFR, FFRRangio, FlashAngio CaFFR, QFR, virtual PCI, pre- and post-PCI angiography-derived FFR

## Abstract

Advances in computer technology and image processing now allow us to obtain from angiographic images a large variety of information on coronary physiology without the use of a guide-wire as a diagnostic information equivalent to FFR and iFR but also information allowing for the performance of a real virtual percutaneous coronary intervention (PCI) and finally the ability to obtain information to optimize the results of PCI. With specific software, it is now possible to have a real upgrading of invasive coronary angiography. In this review, we present the different advances in this field and discuss the future perspectives offered by this technology.

## 1. Introduction 

Invasive coronary angiography (ICA) has long been the gold standard for the diagnosis of coronary artery disease (CAD). However, the poor agreement between the degree of stenosis and the hemodynamic significance of a stenosis has made the anatomical information provided by ICA less important in comparison to invasive coronary physiology. Indeed, ICA can predict the hemodynamic significance of 40–70% coronary stenoses in less than 50% of cases [1]. Invasive coronary physiology has become the reference for the diagnosis of significant epicardial stenosis and is recommended in class I level A by international recommendations to guide revascularization (e.g., fractional flow reserve (FFR), instantaneous wave-free ratio (iFR)) [2,3]. 

However, FFR/iFR remains underused for various reasons. This technique takes time; requires pharmacological induction of hyperemia, which causes patient discomfort; uses pressure guide-wires that are costly; and requires passing through stenosis that can lead to complications such as dissection and embolism. 

Meanwhile, advances in computer technology and image processing now allows us to obtain a large variety of coronary physiology information from angiographic images without using guidewires as diagnostic information equivalent to FFR and iFR. This technology also allows for a virtual percutaneous coronary intervention (PCI) and optimization of PCI results. Specific software makes it possible to upgrade ICA in real time. In this review, we present the different advances in this field and discuss the future perspectives offered by this technology.

## 2. Upgraded ICA for Pre-PCI Time

Advances in computational power have led to the development of angiography-derived FFR, which can be obtained without the need of hyperemia and pressure guidewires. Angiography-derived FFR combines two projections to provide a 3D quantitative coronary angiography, allowing for the reconstruction of a specific coronary geometry. Physiological inputs are used to estimate flow or resistance, and computational fluid dynamics (CFD) techniques or mathematical formulas are used to analyze the pressure drop across a lesion [4,5]. Several angiography-derived FFR software packages have been developed. Quantitative flow ratio (QFR) [6,7,8,9,10], cardiovascular angiographic analysis systems for vessel fractional flow reserve (CAAS vFFR) [11,12,13], and the FFRangio system [14,15,16,17,18,19] are angiographically derived estimates of FFR with comparable performances [20]. The FlashAngio caFFR System is another coronary angiography-derived fractional flow reserve measurement system. The system calculates FFR values based on a three-dimensional (3D) model of the coronary arterial system restored from two angiograms. It differs from other systems in that it uses real-time aortic pressure measurements taken by a specific device (FlashPressure, Rainmed Ltd.). The recorded aortic pressure is averaged over five cardiac cycles and converted to hyperemic aortic pressure using a simple formula. The resting flow velocity is obtained from frame counting, followed by auto-calculation of the hyperemic flow velocity. Finally, a proprietary computational pressure-flow dynamics (CPFD) method is applied to compute the pressure drop across the lesion by solving the Navier–Stokes equation [21]. It is important to note that in order to adapt to the conditions of a Cath-lab, most of the available software do not use CFD to completely solve the Navier–Stokes equations and continuity equations and produce unknown variable pressure. This is because physical properties of the blood and definition of boundary conditions, such as flow velocity in the inlet and outlet of the 3D geometry model, are required for CFD analysis, which is time consuming. Indeed, data analysis initially took ≈24 h [22], and despite subsequent improvements, vFFR data analysis remains an offline process [23]. A fast CDF was developed for QFR, which uses different mathematical equations based upon the laws of Bernoulli and Poiseuille. The boundary conditions include the following assumptions: blood is a homogeneous and Newtonian fluid, coronary pressure is constant in the absence of stenosis, coronary flow velocity is maintained along the coronary, and steady flow is specified as a boundary condition at the outlet. Flow velocity is frame-counted under resting conditions and an assumption/prediction is then made about what the velocity would be if hyperemia was induced. Hyperemic flow velocity is predicted using contrast-QFR (cQFR) based on contrast flow velocity under non-hyperemic conditions by TIMI frame count derived by quadratic function [7]. The amount of pressure drop is determined by the stenosis geometry and the flow moving through it, as described by the fluid dynamic equations [24,25]. For the computation of CAAS vFFR, the microcirculation resistance is first determined by calculating the average relative values of all patients. This value is then applied as a generic condition. Additionally, the resting aortic pressure of the patient during angiography is recorded, and a hyperemic flow rate is empirically estimated by assuming that the flow velocity is preserved along the evaluated vessel. The boundary conditions for the computation include a constant parabolic flow profile at the inlet and a stress-free outlet, as well as a rigid-wall assumption and a Newtonian fluid approximation of blood [11]. The pressure drop is calculated instantaneously by applying physical laws, which consider the viscous resistance and separation loss effects present in coronary flow behavior [24,25]. Using this information, the software computes CASS vFFR, which is defined as the ratio of distal coronary pressure to patient-specific aortic pressure. The characteristics of the FFR flashangio have been previously described. Finally, in contrast to the other angio-derived FFR, FFRangio is based on the 3D reconstruction of the whole coronary artery tree and applies a serial resistance model based on the anatomical features of lesions only, such as length and diameter, to calculate resistance, while neglecting entrance effects and rheology particularities. FFRangio is mainly based on anatomic rather than hemodynamic indices, and a rapid pressure-flow analysis calculates FFRangio defined as the ratio of the maximal flow rate in the stenosed vessel compared with the maximal flow rate in the absence of the stenosis [17]. Table 1 summarizes the characteristics, functioning, and limitations of available applications for angiography-derived FFR determination in clinical practice. 

Angiography-derived FFR has shown excellent performances in diagnosing hemodynamically significant stenoses defined by FFR < 0.80 [5]. Table 2 lists clinical studies evaluating the role of the angiography-derived FFR index and level of correlation with FFR. Angiography-derived FFR improve ICA for pre-PCI time because they allow for the assessment of the severity of coronary stenosis without the disadvantages of FFR. Angiography-derived FFR is faster than FFR, notably because it does not use pharmacological induction of hyperemia or pressure guide-wires. After conducting correlation studies with FFR, several interventional studies are currently underway to evaluate the role of angiography-derived FFR in revascularization. Currently, the quantitative flow ratio (QFR) is the angiography-based index with the largest amount of evidence. The FAVOR III China is a prospective study that included 3825 patients from China in which QFR-guided PCI was used with a cut off ≤0.89 and compared with angiography-guided PCI. The composite endpoint including all-cause death, MI, and ischemia-driven revascularization occurred in 5.8% (11/1913) of patients in the QFR group compared to 8.8% (167/1912) in the angiography group (HR 0.65, 95% CI 0.51–0.83, *p* = 0.004) at 1 year [26]. Further non-inferiority studies against FFR are expected. The FAVOR III EJ study (NCT03729739) is investigating whether QFR-guided PCI will be non-inferior at 12 months compared to an FFR-guided strategy. The FAST III study (NCT0493177) is a randomized controlled, open-label, multicenter, international, non-inferiority strategy trial of a vFFR-guided strategy compared to an FFR-guided strategy to guide coronary revascularization in 2228 subjects with intermediate coronary artery lesions. The endpoint is the rate of the composite of all-cause death, any myocardial infarction, or any revascularization [27]. The Flash FFR II Study (NCT04575207) is a prospective, multicenter, blinded, randomized, noninferiority clinical trial of coronary angiography fractional flow reserve (caFFR) versus fractional flow reserve (FFR) to guide percutaneous coronary intervention with 2132 participants. The endpoint is a composite of all-cause death, myocardial infarction (MI), and unplanned revascularization at 1 year. The PIONEER-IV study (NCT04923191) is a prospective, multicenter, blinded, randomized, non-inferiority clinical trial of coronary angiography fractional flow reserve (caFFR or QFR) versus fractional flow reserve (FFR) to guide percutaneous coronary intervention with 2540 participants. The primary endpoint is a patient-oriented composite endpoint (all-cause death, any stroke, any myocardial infarction, and any clinically and physiologically driven revascularization). The non-inferiority between angiography-derived physiology guidance versus usual care in an all-comers PCI population treated with unrestricted use of the healing-targeted supreme (HT Supreme) drug-eluting stent and P2Y12 inhibitor monotherapy after 1 month of dual-antiplatelet therapy will also be tested. All of these results will be critical in increasing the recommendation rank to guide revascularization of coronary-angiography-derived fractional flow reserve measurement systems. 

## 3. Limits of Technology 

Suboptimal performances of angiography-derived FFR have been reported. It is important to take these limits into account because they will be found in all applications developed in this review. Angiography-derived FFR has shown limitations for lesions located at bifurcations, ostial lesions, or lesions of left main coronary artery. Angiography-derived FFR is also dependent on the quality of angiographic images. It is recommended to center the image and administer glyceryl trinitrate before acquisition in order to have a good catheter engagement to be able to optimize artery contrast opacification, to acquire two optimal angiographic projections >30° acquired at 15 frames/s, to have good visualization of the lesion with minimal overlap or foreshortening of vessels, to acquire over at least four cardiac cycles, to avoid excessive movement of the X-ray tube, to use a zoom that cuts parts of the coronary to be analyzed, and to ensure the ECG signal is captured. Figure 1 shows several steps to measure QFR. To use this technology properly, strict acquisition protocols must be implemented in the Cath-lab. It is also necessary to train the staff to use the software. Interobserver variability remains acceptable (r = 0.87 to 0.96) for all these techniques, providing that the quality of the angiogram is correct and that it is performed by a trained operator [4].

Noninvasive models of angiography-derived FFR therefore rely on assumptions about hyperemic flow and minimal microvascular resistance or predict them from surrogate markers such as arterial diameter. Therefore, angiography-derived FFR is not patient-specific physiology, which can cause problems in some situations (previous myocardial infarction, cardiomyopathy) [42]. Moreover, in the metanalysis by Collet et al., angiography-derived FFR models were not reliable for values ranging between 0.77 and 0.86, suggesting that there is an uncertainty zone that could require additional invasive FFR [20]. 

## 4. Advances in ICA to Assess PCI Results

Several recent studies have contradicted the use of FFR for revascularization [43]. FAME 3 is the study providing the most interesting negative results [44]. In patients with three-vessel coronary artery disease, FFR-guided PCI was not found to be non-inferior to CABG with respect to the incidence of a composite of death, myocardial infarction, stroke, or repeated revascularization at 1 year (10.6% vs. 6.9%; *p* = 0.35 for noninferiority). A post hoc analysis study showed that post-PCI FFR had prognostic value [45]. One of the hypotheses of this failure of PCI may lie in the fact that multiple angioplasties were, in fact, suboptimal. Indeed, many studies with different physiological tools (FFR, iFR, Pd/Pa) have shown that an average of 21–24% of PCIs are sub-optimal, and that these were associated with a significantly worse outcome [46]. Post-PCI physiologic assessment can detect suboptimal PCI with residual ischemia, which is an independent predictor of target vessel failure. Unlike the cut-offs of diagnosis, which are often validated against non-invasive techniques, there is a lack of consensus for post-PCI prognostic cut-offs [47]. This is due to the fact that studies that investigate this aspect do not necessarily include the same populations and have different event rates and definitions of clinical events. Here, we consider that an FFR ≤ 0.86 or an iFR ≤ 0.89 post-angioplasty is pejorative [46]. If it is possible to detect a non-optimal PCI with these indexes, it will be necessary to use additional parameters to identify whether the problem is focal or secondary to a diffuse coronary involvement. It is crucial to investigate the causes of suboptimal post-PCI physiologic results using physiological indexes and/or intravascular imaging. In this setting, visual assessment of the pullback tracing provides a rough idea of the presence of focal or diffuse step-up, but the analysis remains subjective. The hyperemic pullback pressure gradient (PBPG) is a more accurate index [48]. The FFR or IFR gradient per unit of arterial length or per unit of time may also be considered as an additional objective measure, allowing for the differentiation of a diffuse from a focal lesion. For example, a dFFR(t)/dt ≥ 0.035/s or a HPBPG between 0.65 and 1 indicates a focal lesion [49]. Other indexes such as the gain of FFR are useful for prognosis evaluation in diffuse lesions [50]. All of these post-PCI indexes with their cut-offs are summarized in the proposed algorithm in Figure 2. However, FFR and other physiological indexes may be affected due to damage to the coronary microcirculation post-PCI [51]. In post-PCI, non-hyperemic pressure ratios such as iFR can also be influenced by transient changes in boundary conditions such as increased heart rate, elevated sympathetic tone, or post-occlusion reactive hyperemia [47]. All of these elements can disturb post-PCI measurements. Therefore, it is important to allow some time before measuring the results of PCI. However, the time necessary for normalization of coronary hemodynamics is not well known and has not been standardized yet. Moreover, the same reasons that allow the underuse of coronary physiology with guidewires, as well as the need to use duplicated motorized device to perform pullback with a speed of 1 mm/s, will probably lead to underuse of these indexes.

This problem can be circumvented by using angiography-derived FFR, which is free from variations in post-PCI hemodynamic conditions. With angiography-derived FFR, the analysis will not take into account these acute post-PCI hemodynamic disturbances. The measurement will therefore be much more reproducible and will not require time for normalization. The prognostic value of post-PCI QFR was demonstrated in a retrospective study enrolling patients treated with state-of-the-art PCI for de novo three-vessel disease [28]. In the prospective HAWKEYE study, it was shown that a post-PCI QFR ≤ 0.89 was independently related to a poor prognosis. In this study, the rate of suboptimal PCI was 16%, and those were associated with an unfavorable outcome. Among these 16%, subjective analysis of the pullback curve after PCI identified a focal drop inside the stent in 13% of cases and outside the stent in 32%. A total of 34% showed a steady and progressive decrease in the QFR curve, suggestive of diffuse disease, and 21% showed a combination [29]. All invasive coronary physiology indexes used to characterize post-PCI abnormalities have their surrogates derived from angiography [52]. After PCI, the software can generate a QFR pullback curve but also QFR-PBPG and dQFR/ds. A focal lesion is defined by a QFR abrupt pressure drop-down with ΔQFR > 0.05 in <10 mm [53] or QFR-PBPG > 0.78 with major gradient defined by dQFR/dt ≥ 0.005/mm. The residual disease patterns, particularly predominantly diffuse with major gradient, were independently associated with VOCO at 2 years [30]. If suboptimal PCI is found (QFR ≤ 0.89), the causes must be identified by QFR virtual pullback, QFR-PBPG, and dQFR/ds in order to understand whether the problem is focal (such as stent edge dissection or under expansion, or significant stenosis located proximally to the target PCI) and whether an additional stent is required. In such cases, endocoronary imaging such as IVUS or OCT may be used. Figure 3 shows several indexes available after PCI. The amount of data in the PCI item are much less compared with other angiography-derived FFR, with only comparison studies of post-PCI FFR with post-CASS vFFR [31,32].

## 5. Virtual PCI 

Performing a wire-based coronary physiology pullback before PCI can help to identify the distribution and severity of coronary artery disease (CAD) and predict the likely success of the procedure. However, this approach has some limitations. It requires inducing hyperemia, which is time consuming and can be uncomfortable for the patient, as well as involving the use of a device that carries the added risk of passing under a stent mesh when repositioning the guide at the end of the artery. In addition, FFR guides are often less effective than hydrophilic guides for performing angioplasty. More importantly, another issue is that in 15% of cases, operators may decline to perform an intervention to improve the post-PCI outcome [54]. This is because it can be challenging to fully correct suboptimal PCI results. As a result, virtual PCI is becoming a more attractive option. A pre-procedural measurement that can predict the extent to which physiological ischemia will be resolved would be of greater clinical value. This would allow for the selection of the most appropriate devices and enable individualized planning of the procedure.

With the excellent correlation results between FFR and angiography-derived FFR, it was possible to develop the concept of virtual PCI. It is possible to predict the physiological response to a virtual PCI according to the length and diameter of the virtually implanted stent. Several studies have demonstrated that angiography-derived FFR can provide accurate prediction of FFR in comparison with the actual FFR measured after PCI. The design of these studies typically consisted in an analysis of ICA in cohorts of patients who underwent revascularization and FFR measurement before and after PCI. Pre-PCI estimation of “predictive” or “residual” angiography-derived FFR post-PCI based on invasive angiographic imaging is feasible, presents a correct correlation with invasive FFR and angiography-derived FFR measurements after PCI, and is independently associated with a higher risk of 2-year vessel-oriented composite outcome VOCO (vessel-related ischemia-driven revascularization, vessel-related myocardial infarction, or cardiac death) [33,34,35,36]. It is possible to perform a mapping of the entire artery over the course of the virtual PCI. Qualitative and quantitative analyses of the vessel are provided by QFR virtual pullback trace and QFR-PBPG, respectively. Virtual QFR-PBPG is also called the quantitative flow ratio virtual pullback index (QVP). A focal lesion is defined by the qualitative method by the presence of single drop QFR ≥ 0.05 in 10 mm or QFR PBPG > 0.71. A progressive decline of the QFR value without clear evidence of a focal drop or QFR PBPG ≤ 0.51 predicted diffuse patterns. It is interesting that several studies point to a suboptimal PCI result in the setting of diffuse lesions [37]. The QFR-PBPG is a quantitative index that allows for the identification of patients at risk of suboptimal PCI and who presents more events during follow-up, in particular by identifying patients with diffuse lesions [38]. Even after a successful PCI, a pre-PCI QFF-PPBG < 78 (diffuse lesions) signals an excess risk of events [39]. Finally, this classification of patients before PCI can be refined using the dQFR/ds. Four groups of patients can be identified before PCI: predominantly focal disease (QFR-PBPG ≥ 0.78) with (dQFR/dt ≥ 0.025/mm) and without major gradient (dQFR/dt < 0.025/mm) and predominantly diffuse disease (QFR-PBPG < 0.78) with and without major gradient. The cumulative incidence of VOCO after PCI was significantly higher in patients with predominantly diffuse disease [40]. For the virtual PCI, a physiological map created from angio-FFR pullback is also available with Flashangio caFFR. It allows us to generate Δangio-FFR/mm, an equivalent of dQFR/dt index, and a flashangio caFFR-PBPG. These indexes have proven their feasibility for virtual PCI and their prognostic value [41]. This virtual PCI may be of particular interest in cases of tandem stenosis where FFR measurements may be compromised [55].

The predominantly retrospective design of all these studies is regrettable. However, this opens the field to prospective randomized studies to evaluate the prognostic impact of a revascularization strategy guided by virtual PCI. Further studies are needed to evaluate the efficacy and safety of 3D-QCA-based FFR-guided coronary interventions. This virtual PCI is important because it can anticipate a worse post PCI result secondary to a misjudgment of the length of the lesion. Such post-PCI coronary physiology has been associated with a pejorative prognosis and should therefore be anticipated as far as possible. Biscaglia et al. [56] presented results obtained using virtual PCI in a randomized study. The AQVA (angio-based quantitative flow ratio virtual PCI versus conventional angio-guided PCI in the achievement of an optimal post-PCI QFR) trial aimed at testing whether a QFR-based virtual percutaneous coronary intervention (PCI) was superior to a conventional angiography-based PCI at obtaining optimal post-PCI QFR results. Three-hundred patients (356 study vessels) undergoing PCI were randomized 1:1 to receive either QFR-based virtual PCI or angiography-based PCI (standard of care). The primary outcome was the rate of study vessels with a suboptimal post-PCI QFR value, which was defined as <0.90. The primary outcome occurred significantly more frequently in the angiography-based group (*n* = 26, 15.1%) compared with the QFR-based virtual PCI group (*n* = 12, 6.6%; *p* = 0.009) [56]. Figure 4 shows an example of virtual PCI. 

## 6. Future Research

### 6.1. Application to Bifurcations

Software is constantly being improved. It is possible to use this technique in the anatomical setting of bifurcations. This allows for the determination of the angioplasty strategy on the side branch. The system allows for a better analysis of the side branch without having to pass a guide inside. Its accuracy is increased by using the angiography data but also the pullback in OCT of the main branch [57]. The system makes it possible to limit side branch instrumentation and to significantly reduce procedural complexity in order to improve treatment strategy planning and to potentially improve the outcome of bifurcation lesions. The software is actually a three-dimensional (3D) quantitative coronary angiography (QCA) software adapted to the bifurcation, which provides information on three parts of the bifurcation (proximal segment, main and side branches). Accordingly, it allows for a better objectification of the diameter of the branches for the choice of stents as well as for optimization of the angiography view for PCI. The correct assessment of bifurcation lesion anatomy, especially the ostia of branches, is essential in the choice of the treatment strategy. Indeed, meticulous positioning of a side branch stent is of utter most importance to ensure complete ostial lesion coverage and to limit the protrusion of the stent in the main vessel by minimizing the vessel foreshortening and overlap. The 3D QCA allows for the reduction of the volume of contrast medium used and the radiation exposure of the patient and staff [58]. Here again, as seen previously, the software allows us to prepare the PCI strategy before the procedure. The 3D QCA improves the correlations with the FFR measurements by using fractal laws [59]; however, no software allows us to obtain angiography-derived FFR of each branch yet. 

### 6.2. Evaluation of the Coronary Microcirculation Dysfunction 

The impairment of the coronary microcirculation is an important prognosis parameter in chronic as well as acute coronary syndromes [60,61]. Currently, the microcirculatory resistance index (IMR) is the most widely used and tested for the evaluation of coronary microcirculation dysfunction. IMR quantification involves the use of an intracoronary guide wire to estimate microvascular resistance from simultaneous measurements of distal coronary artery pressure and flow by thermodilution during hyperemia [62]. Here again, technological progress has made it possible to use fluid dynamics to obtain an IMR angiogram by means of post-processing software with no need for guidewires or adenosine. Several indexes are available using very similar methodologies, but all of them have demonstrated good correlations with the IMR [63]. Figure 5 shows an IMR angiogram on a patient and its comparison with IMR. Although a correlation has been demonstrated, the validation of the prognostic value of the different angiography-derived IMR is still pending. In an observational study, an Angio-IMR > 40 U by FlashAngio system (FlashAngio, Rainmed Ltd., Suzhou, China) was an independent predictor of cardiac death or readmission for heart failure in STEMI patients (hazard ratio: 2.173; 95% CI: 1.157–4.079; *p* = 0.016) [64]. Several studies are underway using IMR as a theragnostic biomarker to develop stratified medicine [65]. For example, for chronic coronary syndromes, in a DECISIONING study (NCT05178914), patients will benefit from therapeutic escalation or de-escalation according to IMR in comparison with the sham procedure. For STEMI patients, in the RESTORE-AMI study (NCT03998319), only patients with an IMR > 32 will be eligible for randomization between intracoronary tenecteplase and placebo. Similar studies will need to be designed to demonstrate that the use of software to measure angiography-derived IMR can improve prognosis with stratified medicine.

### 6.3. Non-Invasive Diagnostic Method in Coronary Artery Disease

If post-processing imaging improves the performance of ICA, ICA can never be used as a screening test for coronary artery disease. Indeed, ICA remains an invasive procedure with its own risks, including death, and this strategy is not cost effective when compared to strategy using a non-invasive diagnostic method in coronary artery disease [66]. Many methods exist, such as stress echocardiography, single-photon emission computed tomography, positron emission tomography (PET), stress prefusion cardiac magnetic resonance imaging (CMR), and FFR-CT. Several meta-analyses show superior diagnostic performance with FFR as a reference for CMR and PET [67,68]. The FFR-CT method, which we discuss below for the development of a virtual PCI, had a low positive predictive value in a very recent study in a real-world multicenter with cost analysis. Thus, FFR-CT use is potentially more expensive than conventional stress imaging strategies to detect coronary artery disease [69].

### 6.4. Development of Pre-PCI via Coronary Computed Tomography Angiography (CCTA)

In the years to come, all of the above elements could be available via CCTA. As with virtual PCI using ICA analysis systems, here it will be possible to predict PCI results from CCTA images and the HeartFlow^®^ analysis. The comparison with the data predicted on the whole vessel with the pullback FFR and pullback OCT performed after PCI provides excellent results. Actual post-PCI FFR was 0.88 ± 0.06, and the FFRCT Planner FFR was 0.86 ± 0.06 (mean difference: 0.02 ± 0.07 FFR unit; limits of agreement: −0.12 to 0.15). Minimal stent area by optical coherence tomography was 5.60 ± 2.01 mm^2^, and FFRCT Planner minimal stent area was 5.0 ± 2.2 mm^2^ (mean difference: 0.66 ± 1.21 mm^2^; limits of agreement: −1.7 to 3.0) [70]. The accumulation of anatomic information, especially from the vessel wall, represents an advantage for interventional planning by assessing the calcium score and distribution of calcified plaque, providing a detailed assessment of plaque extension, volume, and composition, as well as an identification of coronary lesions requiring additional interventions (e.g., atherectomy) to improve stent implantation success [71].

CCTA analysis is often criticized for being lengthy—this will probably be improved, but in the context of CCTA, which is performed upstream of ICA, this delay may be acceptable. In the near future, it is certain that CCTA and ICA information will be available simultaneously during the procedure in order to optimize revascularization. A software prototype already exists for integration in Cath-lab that reconstructs the vessel lumen and atherosclerotic plaques from the CCTA information and projects them onto the 2D images of invasive angiography. The movement of the C-arm is tracked in real time by an external sensor to synchronize the orientation of the 3D coronary tree of CCTA with the projection of the fluoroscopic C-arm [72]. 

This technology could be very interesting in the Cath-lab for PCI guidance in the field of CTO, allowing for enhanced visualization of the occluded vessel (providing an online 3D-vessel view with a characterization of proximal and distal cap, length of lesion, and spatial orientation) and leading to optimized strategies for recanalization [73].

## 7. Conclusions 

Due to the progress in imaging post-processing, ICA is an examination that now allows both anatomical and physiological analysis of the coronary tree. The widespread use of these devices will allow for the conduct of randomized studies focused on clinical outcomes. Such studies represent an essential step in the validation of these new technologies in clinical practice.

## Figures and Tables

**Figure 1 diagnostics-13-01978-f001:**
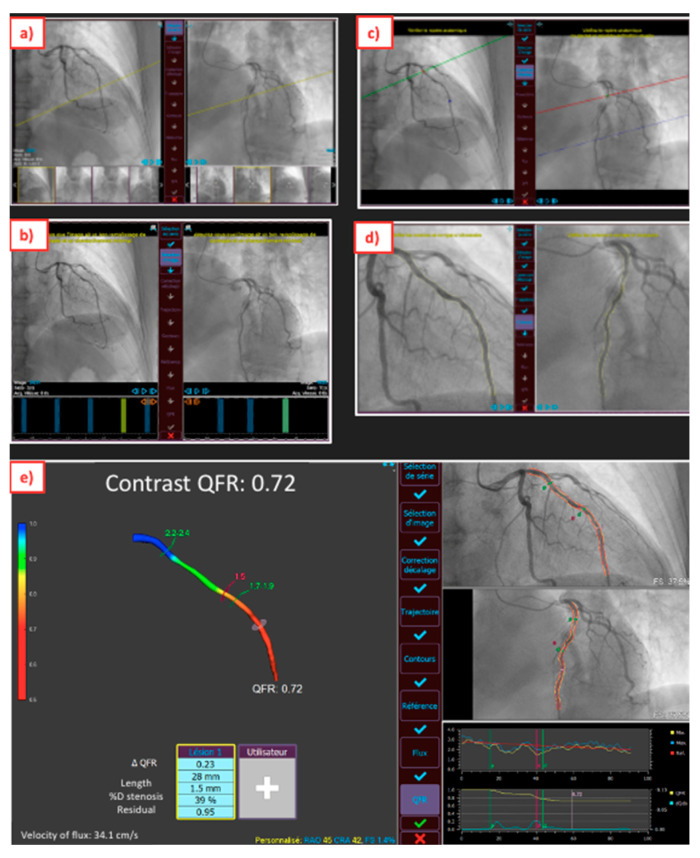
Example of the steps of the QAngio XA 3D software (research edition, version 2.0, Medis, Leiden, the Netherlands) allowing angiography-derived FFR. The reconstruction consisted of the following steps: (**a**) selection of 2 angiographic image sequences >25° apart; (**b**) selection of proper contrast-filled end-diastolic frames; (**c**) identification of anatomical landmarks for automated correction of system distortions; (**d**) delineation of the bifurcation lumen in the 2 projections; (**e**) 3D reconstruction of the lumen and the reference surface with QFR value.

**Figure 2 diagnostics-13-01978-f002:**
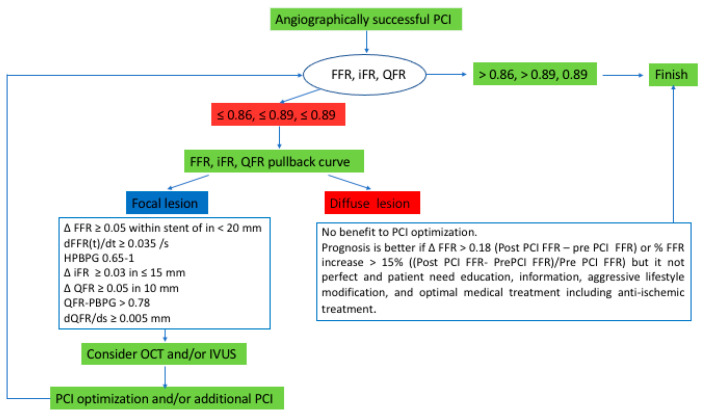
Proposal of an algorithm for post-PCI optimization.

**Figure 3 diagnostics-13-01978-f003:**
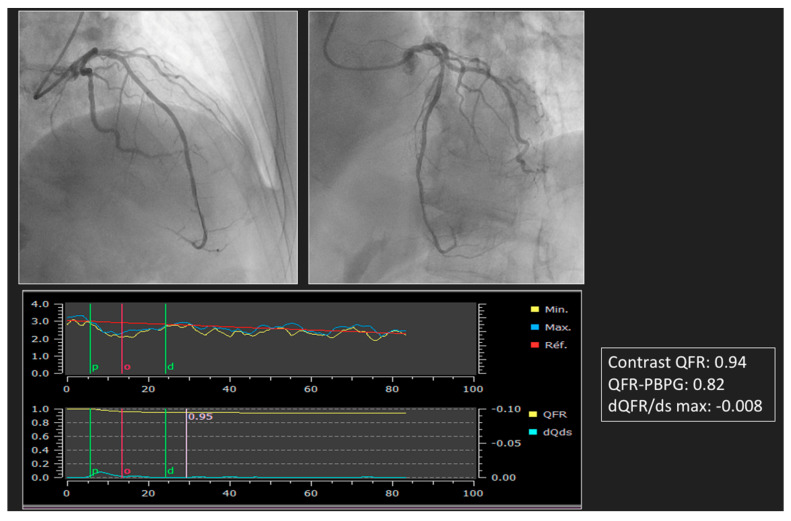
Post-PCI index derived by angiography (post-PCI QFR, pullback QFR, QFR-PPG, dQFR/ds).

**Figure 4 diagnostics-13-01978-f004:**
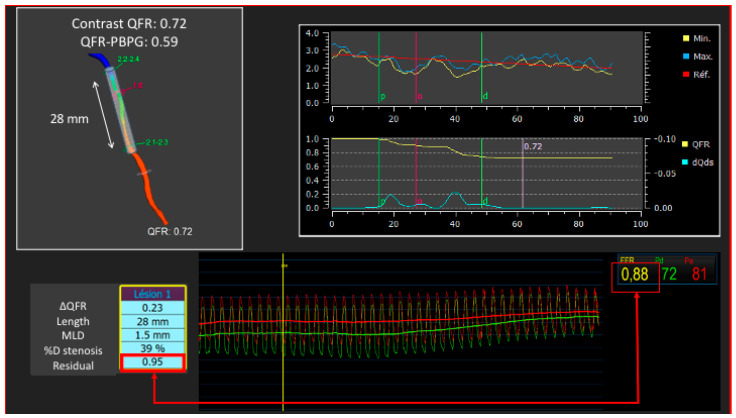
Virtual PCI planned before PCI and comparison with post-PCI measure by guide-wire pressure and angiography-derived measure.

**Figure 5 diagnostics-13-01978-f005:**
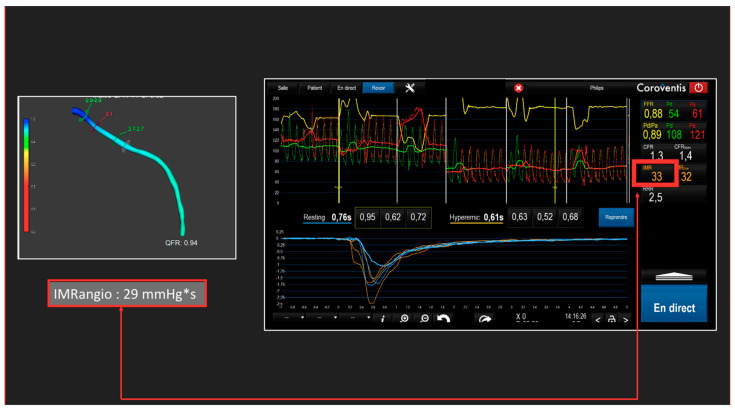
Comparison between measure of IMR by thermodilution and angiography-derived IMR by QAngio XA 3D software (research edition, version 2.0, Medis, Leiden, the Netherlands) in a patient after PCI.

**Table 1 diagnostics-13-01978-t001:** Characteristics of available angiography-derived FFR applications available for clinical practice.

	Index			
	QFR	FFRangio	CASS vFFR	FlashAngio caFFR
**Vendors**	Medis Medical Imaging	CathWorks	Pie Medical Imaging	RainMed
**Angiography**	2 projections > 25° apart	≥2 projections 30° apart	2 projections > 30° apart	2 projections > 30° apart
**Data inputs**	-3D coronary model -TIMI frame counting	-3D coronary model -Aortic pressure	-3D coronary model -Empiric hyperemic flow -Aortic pressure	-3D coronary model -TIMI frame counting -Dynamic aortic pressure
**Post processing**	Integrated mathematical approach (Bernoulli and Poiseuille)	Flow resistance analysis	Integrated mathematical approach (Bernoulli and Poiseuille)	Computational pressure-flow dynamics
**Number of vessels analyzed**	1	Multi-vessels	1	1
**Cut-off value**	0.8	0.8	0.8	0.8
**Processing time**	5 min	3.41 min	NA	4.5 min
**Limitations**	-Correct angiogram is necessary	-Correct angiogram is necessary -Model is mainly based on anatomical features	-Correct angiogram is necessary but even with that high rate of excluded images	-Correct angiogram is necessary with use of auto-injector -Few proofs

CASS vFFR: cardiovascular angiographic analysis systems for vessel fractional flow reserve, FFRRangio: fractional flow reserve derived from invasive angiography, FlashAngio CaFFR: flashangio computational pressure-flow-dynamics-derived FFR, QFR: quantitative flow ratio.

**Table 2 diagnostics-13-01978-t002:** Clinical studies evaluating the role of angiography-derived FFR indexes.

First Authors, Year, (Ref. #)	*N* of Patients	Index	Study Methods	Results
Pre PCI				
Masdjedi et al. [11], 2020	100	CASS vFFR	Correlation ofCASS vFFR and FFR in a retrospective study	CASS vFFR correlated with FFR (r = 0.89, *p* < 0.001), with 95% limit of agreement ± 0.05
Neleman et al. [12], 2021	912	CASS vFFR	Correlation of CASS vFFR and FFR in a retrospective study	CASS vFFR correlated with FFR (r = 0.89, *p* < 0.001)
Masdjedi et al. [13], 2022	334	CASS vFFR	Correlation of CASS vFFR and FFR in a prospective study	CASS vFFR correlated with FFR (r = 0.74, *p* < 0.001), with 95% limit of agreement ± 0.12
Kornowski et al. [14], 2016	88	FFRangio	Correlation of FFRangio and FFR in a prospective study	FFRangio correlated with FFR (r = 0.90, *p* < 0.001)
Kornowski et al. [15], 2018	53	FFRangio	Correlation of FFRangio and FFR in a prospective study	FFRangio correlated with FFR (r = 0.91, *p* < 0.001), with 95% limit of agreement ± 0.07
Pellicano et al. [16], 2017	184	FFRangio	Correlation of FFRangio and FFR in a prospective study	FFRangio correlated with FFR (r = 0.88, *p* < 0.001), with 95% limit of agreement ± 0.10
Fearon et al. [17], 2019	301	FFRangio	Correlation of FFRangio and FFR in a prospective study	FFRangio correlated with FFR (r = 0.80, *p* < 0.001), with 95% limit of agreement ± 0.13
Omori et al. [18], 2019	50	FFRangio	Correlation of FFRangio and FFR in a prospective study	FFRangio correlated with FFR (r = 0.83, *p* < 0.001), with 95% limit of agreement ± 0.13
Witberg et al. [19], 2020	588	FFRangio	Correlation of FFRangio and FFR; a pooled analysis of 5 prospective cohort studies	FFRangio correlated with FFR (r = 0.83, *p* < 0.001), with 95% limit of agreement ± 0.11
Tu et al. [6], 2014	68	QFR	Correlation of QFR and FFR in a retrospective study	QFR correlated with FFR (r = 0.81, *p* < 0.001), with 95% limit of agreement ± 0.11
Tu et al. [7], 2016	73	QFR	Correlation of QFR and FFR in a prospective study	QFR correlated with FFR (r = 0.77, *p* < 0.001), with 95% limit of agreement ± 0.12
Westra et al. [8], 2018	191	QFR	Correlation of QFR and FFR in a prospective study	QFR correlated with FFR (r = 0.70, *p* < 0.0001), with 95% limit of agreement ± 0.16
Westra et al. [9], 2018	272	QFR	Correlation of QFR and FFR in a prospective study	QFR correlated with FFR (r = 0.83, *p* < 0.001), with 95% limit of agreement ± 0.12
Stähli et al. [10], 2019	436	QFR	Correlation of QFR and FFR in a retrospective study	QFR correlated with FFR (r = 0.8, *p* < 0.001), with 95% limit of agreement ± 0.07
Xu et al. [26], 2021	3847	QFR	Multicenter, blinded randomized between a QFR-guided strategy or an angiography-guided strategy; the primary endpoint was the 1-year rate of major adverse cardiac events, a composite of death from any cause, myocardial infarction, or ischemia-driven revascularization	The 1-year primary endpoint occurred in 5.8% in the QFR-guided group and in 8.8% participants in the angiography-guided group (*p* = 0·0004)
Li et al. [21], 2020	328	FlashAngio caFFR	Correlation of FlashAngio caFFR and FFR in a prospective multicenter study	FlashAngio caFFR correlated with FFR (r = 0.89, *p* < 0.001), with a 95% limit of agreement ± 0.09
Post PCI				
Kogame et al. [28], 2019	393	QFR	Prognostic value of QFR measured immediately after PCI in patients with a de novo 3-vessel disease-retrospective study by VOCO at 2 years	The incidence of 2-year VOCO in the vessels with post-PCI QFR < 0.91 was significantly higher compared with vessels with post-PCI QFR ≥ 0.91 (12.0% vs. 3.7%; HR: 3.37; 95% confidence interval: 1.91 to 5.97; *p* < 0.001).
Biscaglia et al. [29], 2019	602	QFR	Prognostic value of QFR measured immediately after a PCI-prospective study	ROC analysis identified a post-PCI QFR best cutoff of ≤ 0.89 (AUC 0.77; 95% confidence interval: 0.74 to 0.80; *p* < 0.001). Post-PCI QFR ≤ 0.89 was independent associated with VOCO (hazard ratio: 2.91; 95% confidence interval: 1.63 to 5.19; *p* < 0.001).
Dai et al. [30], 2022	1335	QFR-PBPG, dQFR/ds	Prognosis of 4 patient groups defined by predominant focal disease (QFR-PBPG > 0.78) with (dQFR/dt ≥ 0.005/mm) and without major gradient and predominant diffuse disease (QFR-PBPG ≤ 0.78) with and without major gradient by VOCO in a 2-year retrospective study	At 2 years, VOCO was lowest in patients with predominant focal without major gradient (1.4% vs. 5.4% in predominant focal with major gradient patients vs. 4.8% in predominant diffuse without major gradient patients vs. 8.5% in predominant diffuse with major gradient patients, all *p* < 0.05), whereas there was no prognostic value for classifications by visual assessment. Physiological residual disease patterns were independently associated with VOCO and showed increased prognostic value when introduced to a model with clinical risk factors only (C index: 0.77 vs. 0.68, *p* < 0.008; NRI: 0.65, *p* < 0.001; IDI: 0.020, *p* < 0.001).
Masdjedi et al. [31], 2020	100	CASS vFFR	Correlation CASS vFFR and a FFR-retrospective study	CASS vFFR correlated with FFR (r = 0.88, *p* < 0.001), with 95% limit of agreement ± 0.06
Pizzato et al. [32], 2020	115	CASS vFFR	Correlation CASS vFFR and FFR pre and post PCI-retrospective study	-CASS vFFR could be analyzed in about one-third of previously completed angiographies. -Pearson’s correlation coefficient between pre-PCI FFR and CASS vFFR was 0.449 (*p* = 0.0001). -Pearson’s correlation coefficient between post-PCI FFR and CASS vFRR was 0.115 (*p* = 0.2703).
Virtual PCI				
Rubimbura et al. [33] 2020	93	QFR	Residual QFR and post-PCI QFR were compared to a post-PCI FFR-retrospective study	The correlation coefficient of residual QFR with post-PCI FFR was 0.68 (95% CI: 0.53–0.78) and the correlation coefficient of post-PCI-QFR with post-PCI FFR was 0.79 (95% CI: 0.70–0.86).
Lee et al. [34], 2022	274	QFR	QFR and residual QFR were compered to pre-PCI FFR and post-PCI FFR. Prognosis of rQFR was analyzed by a VOCO-retrospective study.	-Pre-PCI QFR and FFR were correlated (r = 0.756, *p* < 0.001). -rQFR and FFR post PCI were correlated (r = 0.528, *p* < 0.001). -rQFR predicted incidence of 2-year VOCO after index PCI (AUC: 0.712 [0.555–0.869], *p* = 0.041).
Tomaniak et al. [35], 2022	81	CASS vFFR	Residual CASS vFFR and post-PCI CASS vFFR were compared to a post-PCI FFR-retrospective study.	-Residual vFFR and post-PCI FFR were correlated (r = 0.84, *p* < 0.001). -Residual CASS vFFR and post-PCI CASS vFFR were correlated (r = 0.77, *p* < 0.001).
Zhang et al. [36], 2022	2348	QFR	Concordance between residual QFR and post-PCI QFR; prognostic value of residual QFR (VOCO); and forecast of outcomes by virtual randomized controlled trials between residual QFR and an angiographic guidance- retrospective study.	Residual QFR and post-PCI QFR were correlated (*r* =0.976, *p* < 0.0001). Low residual QFR (≤0.92) was independently associated with higher risk of 2-year VOCO (adjusted hazard ratio: 5.50; 95% confidence interval: 3.03 to 10.0). Simulated residual QFR-guided strategy had a 2.6% absolute reduction of 2-year incidence of VOCO compared with the angiography-guided strategy.
Biscaglia et al. [37], 2022	111	Pullback QFR, QFR-PBPG	Analyze the link between focal, serial lesions; diffuse disease; combination defined by pullback QFR or QFR PBPG prePCI with suboptimal PCI result (post-PCI QFR value ≤ 0.89) in post hoc analysis of the HAWKEYE study.	Suboptimal PCI result occurrences differed across functional patterns of CAD (focal 8% vs. serial lesions 15% vs. diffuse disease 33% vs. combination 29%, *p* = 0.03). Similarly, QFR-PBPG was correlated with post-PCI QFR value (r = 0.62, 95% CI 0.50–0.72).
Dai et al. [38], 2022	1003	QFR-PBPG	Prognostic value QFR-PBPG prePCI by VOCO in a 2-year retrospective study.	After multiple adjustment, QFR-PBPG was an independent predictor for VOCO (HR 1.30, 95% CI 1.05–1.62). The addition of QFR-PBPG to the model of clinical risk factors substantially improved the predictions of VOCO (C-index 0.67 vs. 0.62, net reclassification index 0.42).
Dai et al., [39], 2022	1744	QFR-PBPG	Prognostic value of classification according to dichotomous pre-PCI QFR-PBPG and post-PCI QFR by VOCO in a 2-year retrospective study.	Vessels with low pre-PCI QFR-PBPG (3.9% versus 2.0%, HR 1.93; 95% CI, 1.08–3.44; *p* = 0.02) or low post-PCI QFR-PBPG (9.8% versus 2.7%, HR, 3.78; 95% CI, 1.61–8.87; *p* = 0.001) demonstrated higher VOCO risk after PCI. Despite high post-PCI QFR-PBPG being achieved, vessels with low pre-PCI QFR-PBPG presented a higher risk of VOCO than those with high pre-PCI QFR-PBPG (3.7% versus 1.8%, HR, 2.03; 95% CI, 1.09–3.76; *p* = 0.03).
Shin et al. [40], 2021	341	QFR-PBPG, qQFR/ds	Prognosis of 4 patient groups defined by predominant focal disease (QFR-PBPG ≥ 0.78) with (dQFR/dt ≥ 0.025/mm) and without major gradient and predominant diffuse disease (QFR-PBPG < 0.78) with and without major gradient by VOCO in a 2-year retrospective study.	Cumulative incidence of VOCO after PCI was significantly higher in patients with predominant diffuse disease (8.1% predominant diffuse disease with major gradient and 9.9% in predominant diffuse disease without major gradient vs. 1.4% predominant focal disease with major gradient and 0.0% in predominant focal disease without major gradient; overall *p* = 0.024).
Dai et al. [41], 2023	286	Flashangio ca FFR	Residual flashangio caFFR and post-PCI flashangio caFFR were compared to post-PCI FFR, and the prognosis value of residual flashangio caFFR was analyzed in a retrospective study (VOCO).	-Pre-PCI flashangio caFFR (r = 0.88, *p* < 0.001) and post-PCI flashangio caFFR (r = 0.76, *p* < 0.001) showed a close correlation with FFR. -Residual flashangio caFFR and post PCI FFR were correlated (r = 0.88, *p* < 0.001). Residual flashangio caFFR and post-PCI flashangio caFFR were correlated (r = 0.82, *p* < 0.001). -Suboptimal residual flashangio caFFR (≤0.89) was associated with increased risk of 2-year VOCO (adjusted HR: 3.71; 95% confidence interval: 1.50−9.17).

VOCO: vessel-oriented composite outcome (vessel-related ischemia-driven revascularization, vessel-related myocardial infarction, or cardiac death), PBPG: pullback pressure gradient. Same abbreviations as Table 1.

## Data Availability

Not applicable.

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
