# Peer review of "Coronary Angiography Upgraded by Imaging Post-Processing: Present and Future Directions"

_diagnostics, 2023, doi:10.3390/diagnostics13111978_

Round 1
Reviewer 1 Report
The review entitled “Coronary Angiography up-graded by Imaging Post Processing: Present and Future Directions” by Caullery et al adequately summarizes current evidence on imaging post-processing tools of coronary angiography. In particular, the authors describe three possible applications of these tools:
- evaluation of physiological severity of coronary stenosis;
- evaluation of PCI result (physiological assessment after stents deployment);
- method to conduct virtual PCI.
Primary evidences in this field have been included, and ongoing studies regarding these tools have been cited in the manuscript. Moreover, the authors reported on future applications of these post-processing imaging techniques, such as PCI of bifurcation and evaluation of coronary microcirculation dysfunction, for which few evidences are now available.
The work is well written and conducted, only few changes are needed.
1) The second section of the manuscript (Upgraded ICA for pre-PCI time), is full of information and unclear. I would divide this section into two parts as follow:
- first paragraph: a brief description of the four angiography-derived FFR software packages, including their functioning and their limitations;
- second paragraph: Upgraded ICA for pre-PCI time, use of these tools in evaluating coronary stenosis severity, current evidence, and future prospective.
2) Section 4, page 11, lines 15 to 16: “To give some indicators we will consider an FFR < 0.86 or an iFR < 0.89 post-angioplasty may be considered pejorative”. Please correct this sentence.
3) Section 5, page 13, lines 16 to 18: “It is be possible to predict the physiological response to a virtual PCI according to length and diameter stent virtually implanted on the pathological artery”. Please correct this sentence.
4) Section 6: please consider including a brief description of the future role of CCTA in PCI planning of CTO lesions (https://www.jacc.org/doi/full/10.1016/j.jcin.2023.03.013)
Author Response
Response to Reviewers
Thank you very much for the work of the reviewers and editors, which will increase the interest of the manuscript. You will find below a point-by-point answer to each of the reviewers’ and editor’s comments and the indications of changes made to the manuscript.
Reviewers' comments:
Review 1:
The review entitled “Coronary Angiography up-graded by Imaging Post Processing: Present and Future Directions” by Caullery et al adequately summarizes current evidence on imaging post-processing tools of coronary angiography. In particular, the authors describe three possible applications of these tools:
- evaluation of physiological severity of coronary stenosis;
- evaluation of PCI result (physiological assessment after stents deployment);
- method to conduct virtual PCI.
Primary evidences in this field have been included, and ongoing studies regarding these tools have been cited in the manuscript. Moreover, the authors reported on future applications of these post-processing imaging techniques, such as PCI of bifurcation and evaluation of coronary microcirculation dysfunction, for which few evidences are now available.
The work is well written and conducted, only few changes are needed.
1) The second section of the manuscript (Upgraded ICA for pre-PCI time), is full of information and unclear. I would divide this section into two parts as follow:
- first paragraph: a brief description of the four angiography-derived FFR software packages, including their functioning and their limitations;
- second paragraph: Upgraded ICA for pre-PCI time, use of these tools in evaluating coronary stenosis severity, current evidence, and future prospective.
For clarity, we have reorganized the second section into 2 paragraphs as advised.
2) Section 4, page 11, lines 15 to 16: “To give some indicators we will consider an FFR < 0.86 or an iFR < 0.89 post-angioplasty may be considered pejorative”. Please correct this sentence.
We have corrected this sentence: “Here, we will consider that an FFR < 0.86 or an iFR < 0.89 post-angioplasty is pejorative”.
3) Section 5, page 13, lines 16 to 18: “It is be possible to predict the physiological response to a virtual PCI according to length and diameter stent virtually implanted on the pathological artery”. Please correct this sentence.
We have corrected this sentence: “It is possible to predict the physiological response to a virtual PCI according to the length and diameter of the virtually implanted stent”.
4) Section 6: please consider including a brief description of the future role of CCTA in PCI planning of CTO lesions (https://www.jacc.org/doi/full/10.1016/j.jcin.2023.03.013)
We have added this description: “This technology could be very interesting in the cath lab for PCI guidance in field of CTO allowing enhanced visualization of the occluded vessel (providing an online 3D-vessel view with characterization of proximal and distal cap, length of lesion and spatial orientation) and leading to optimized strategies for recanalization. (Poletti E, Ohashi H, Sonck J, Castaldi G, Benedetti A, Collet C, Agostoni P, Zivelonghi C. Coronary CT-Guided Minimalistic Hybrid Approach for Percutaneous Chronic Total Occlusion Recanalization. JACC Cardiovasc Interv. 2023 May 8;16(9):1107-1108).
Reviewer 2 Report
In this paper the authors I have presented a comprehensive review of technology in the space of Andrew graphically derived physiology to help guide PCI. I find the paper quite thorough and interesting. The table summarizing the clinical trial results is particularly helpful in evaluating just how far this field has come.
As pointed out by the authors there are limitations with this technology, especially related to assumptions about coronary resistance hyperemic flow, and microvascular function. Probably worth looking at whether this technology (angio FFR) is less accurate in diabetics, for example.
On page 10, I would suggest changing the heading title:
Currently - 4. Up-graded of ICA for Post PCI time page 10
Change to, 4. Advances in ICA to Assess PCI Results
Overall, the English is surprisingly good. Some minor edits and review suggested.
Author Response
Response to Reviewers
Thank you very much for the work of the reviewers and editors, which will increase the interest of the manuscript. You will find below a point-by-point answer to each of the reviewers’ and editor’s comments and the indications of changes made to the manuscript.
Reviewers' comments
Comments and Suggestions for Authors
In this paper the authors I have presented a comprehensive review of technology in the space of Andrew graphically derived physiology to help guide PCI. I find the paper quite thorough and interesting. The table summarizing the clinical trial results is particularly helpful in evaluating just how far this field has come.
As pointed out by the authors there are limitations with this technology, especially related to assumptions about coronary resistance hyperemic flow, and microvascular function. Probably worth looking at whether this technology (angio FFR) is less accurate in diabetics, for example.
On page 10, I would suggest changing the heading title:
Currently - 4. Up-graded of ICA for Post PCI time page 10
Change to, 4. Advances in ICA to Assess PCI Results
We have made this change.
Comments on the Quality of English Language
Overall, the English is surprisingly good. Some minor edits and review suggested.
We have corrected minor English mistakes.
Reviewer 3 Report
The authors reviewd the different advances using the information of coronary physiology without the use of guide-wire as diagnostic information equivalent to FFR and iFR but also information allowing to perform a real virtual Percutaneous Coronary Intervention (PCI) and finally to obtain information to optimize the results of PCI.
This is a very nice e timely revision of literature data. I have only a suggestion to write a subheading ragarding an non-invasive diagnostic methods in cooronary artery disease (please cite: DOI: 10.1186/1476-7120-6-21 )
Minor english mistakes
Author Response
Response to Reviewers
Thank you very much for the work of the reviewers and editors, which will increase the interest of the manuscript. You will find below a point-by-point answer to each of the reviewers’ and editor’s comments and the indications of changes made to the manuscript.
Reviewers' comments
Review 3:
The authors reviewed the different advances using the information of coronary physiology without the use of guide-wire as diagnostic information equivalent to FFR and iFR but also information allowing to perform a real virtual Percutaneous Coronary Intervention (PCI) and finally to obtain information to optimize the results of PCI.
This is a very nice timely revision of literature data. I have only a suggestion to write a subheading regarding a non-invasive diagnostic method in coronary artery disease (please cite: DOI: 10.1186/1476-7120-6-21)
We have written a session on noninvasive diagnostic method in coronary artery diseas:
” 6.3 Non-invasive diagnostic method in coronary artery disease
If post-processing imaging improves the performance of ICA, ICA can never be used as a screening test for coronary artery disease. Indeed, ICA remains an invasive procedure with its own risks, including death and this strategy is not cost effective when compared to strategy using non-invasive diagnostic method in coronary artery disease (Bedetti G, Pasanisi EM, Pizzi C, Turchetti G, Loré C. Economic analysis including long-term risks and costs of alternative diagnostic strategies to evaluate patients with chest pain. Cardiovasc Ultrasound. 2008 May 29;6:21). Many methods exist such as stress echocardiography, single-photon emission computed tomography, positron emission tomography (PET), stress prefusion cardiac magnetic resonance imaging (CMR) and FFR-CT. Several meta-analyses show superior diagnostic performance with FFR as a reference for CMR and PET. (Takx RA, Blomberg BA, El Aidi H, Habets J, de Jong PA, Nagel E, Hoffmann U, Leiner T. Diagnostic accuracy of stress myocardial perfusion imaging compared to invasive coronary angiography with fractional flow reserve meta-analysis. Circ Cardiovasc Imaging. 2015 Jan;8(1):e002666) / Danad I, Szymonifka J, Twisk JWR, Norgaard BL, Zarins CK, Knaapen P, Min JK. Diagnostic performance of cardiac imaging methods to diagnose ischaemia-causing coronary artery disease when directly compared with fractional flow reserve as a reference standard: a meta-analysis. Eur Heart J. 2017 Apr 1;38(13):991-998). The FFR-CT method, which we will discuss below for the development of a virtual PCI, it had a low positive predictive value in a very recent study in real-world multicenter with cost Analysis. Then, FFR-CT use is potentially more expensive than conventional stress imaging strategies to detect coronary artery disease (Mittal TK, Hothi SS, Venugopal V, Taleyratne J, O'Brien D, Adnan K, Sehmi J, Daskalopoulos G, Deshpande A, Elfawal S, Sharma V, Shahin RA, Yuan M, Schlosshan D, Walker A, Abdel Rahman SE, Sunderji I, Wagh S, Chow J, Masood M, Sharma S, Agrawal S, Duraikannu C, McAlindon E, Mirsadraee S, Nicol ED, Kelion AD. The Use and Efficacy of FFR-CT: Real-World Multicenter Audit of Clinical Data With Cost Analysis. JACC Cardiovasc Imaging. 2023 Mar 9:S1936-878X(23)00099-2)”.
Comments on the Quality of English Language
Minor English mistakes
We have corrected minor English mistakes.